# Equity Market Contagion in Return Volatility during Euro Zone and Global Financial Crises: Evidence from FIMACH Model

**A. M. M. Shahiduzzaman Quoreshi * , Reaz Uddin * and Viroj Jienwatcharamongkhol**

Department of Industrial Economics, Blekinge Institute of Technology, SE-371 79 Karlskrona, Sweden; viroj.jienwatcharamongkhol@bth.se

* Correspondence: shahiduzzaman.quoreshi@bth.se (A.M.M.S.Q.); reaz_uddin@yahoo.com (R.U.)

**Abstract:** The current paper studies equity markets for the contagion of squared index returns as a proxy for stock market volatility, which has not been studied earlier. The study examines squared stock index returns of equity in 35 markets, including the US, UK, Euro Zone and BRICS (Brazil, Russia, India, China and South Africa) countries, as a proxy for the measurement of volatility. Results from the conditional heteroskedasticity long memory model show the evidence of long memory in the squared stock returns of all 35 stock indices studied. Empirical findings show the evidence of contagion during the global financial crisis (GFC) and Euro Zone crisis (EZC). The intensity of contagion varies depending on its sources. This implies that the effects of shocks are not symmetric and may have led to some structural changes. The effect of contagion is also studied by decomposing the level series into explained and unexplained behaviors.

**Keywords:** contagion; financial markets; global financial crisis; Euro zone crisis; long memory

**JEL Classification:** C13; C22; C25; C51; G12; G14

## 1. Introduction

The US subprime crisis, also referred to as the global financial crisis (GFC), in 2008 and the eventual Euro Zone crisis (EZC) beginning in 2009 are the most devastating financial crises in recent history. The collapse of Bear Stearns in early 2008 was the prelude for the GFC. Lehman Brothers going bankrupt, Merrill Lynch being taken over by the Bank of America, and the bailout of AIG signals in September 2008 marked the most critical point in the crisis. By the end of 2009, European economies start to fall into debt crises by varying degrees. Notably, Greece, Italy, Ireland, Spain, and Portugal are hard hit; Greece being the worst affected since the crisis hit the Euro Zone in 2010. Academia considers the GFC and EZC as the period of deepest recession in the post-World War II economic order.

The financial market contagion (i.e., increased correlation between stock markets) is an extensively researched subject (e.g., Caporale et al. 2005; Forbes and Rigobon 2002; Mollah et al. 2016). The case of contagion is examined in empirical studies of the 1987 crash of the US stock market, the GFC, EZC, as well as Mexican, Brazilian, Russian and Asian crises. Authors use different sample sizes, period and nature of markets with results leading to more than one conclusion. Comparing data from developed markets, King and Wadhwani (1990) find a significant increase in correlations between the US and UK and other equity markets after the 1987 crash. Lee and Kim (1993) not only confirm the contagion of the 1987 crash, but also show its extent beyond the developed markets incorporating the analysis of emerging markets. Calvo and Reinhart (1996) also document increased correlations among emerging markets in their analysis of the 1994 Mexican crisis. However, contrary to Lee and Kim's (1993) findings,

Forbes and Rigobon (2002) conclude against contagion despite the interdependence in both cases in their study of Mexican and Asian crises of 1994 and 1997, respectively, among 24 countries, including developed and emerging economies. Nonetheless, applying a longer sample period Chiang et al. (2007) show contagion during the Asian crisis of 1994 and 1997. Baig and Goldfajn (1998) study the Asian currency crisis and also find a contagion effect between the equity market and currency market. Caporale et al. (2005) studied the Asian crisis and corroborated the presence of contagion as they found an important increase in co-movements in sampled South-East Asian countries. Corsetti et al. (2005) investigated seventeen developed and emerging countries. The findings are not as conclusive as most of the above studies, showing contagion for less than one-third of the sample countries and tends to be closer to those of Forbes and Rigobon (2002). Similar results are found also in a significant number of recent studies (e.g., Wang et al. 2017; Gamba-Santamaria et al. 2017; Jiang et al. 2017; Bonga-Bonga 2018). Wang et al. (2017) find evidence for contagion during the GFC in G7 countries (except for Japan), Russia and India where the US is used as a source of contagion, and no contagion is found in Brazil, China, and Japan from the same source. Jiang et al. (2017) note that the correlation of stock markets between the US, Britain, Germany, Japan and Hong Kong increases markedly after the crisis, while it exhibits a reverse trend with the Chinese stock market.

The current paper studies equity markets for the contagion of return volatilities under the framework of long memory. Squared stock index returns of equity in 35 markets including the USA, UK, Euro Zone, and BRICS countries are studied as a proxy for the measurement of volatility. A significant number of studies are conducted on the spurious long memory in different context (e.g., Granger and Hyung 1999; Engle and Smith 1999; Diebold and Inoue 2001). However, Bhardwaj and Swanson (2006) find, in line with previous studies (e.g., Granger and Ding 1996), evidence of long memory in squared returns, absolute returns and log-squared returns. Hence, a long memory model for empirical study is called for.

Granger (1980), Granger and Joyeux (1980) and Hosking (1981) formulate autoregressive fractionally integrated moving average (ARFIMA) models in order to account for long memory property. Interestingly, significant time elapsed before Bhardwaj and Swanson (2006) conducted an empirical study focusing on the usefulness of the ARFIMA model. They find convincing evidence to apply ARFIMA in squared, log-squared, and absolute stock index returns. Meanwhile, Granger and Ding (1996) identify other processes which also would demonstrate the long-memory property, while Baillie et al. (1996) and Chung (1999) develop and modify a fractionally integrated generalized autoregressive conditional heteroskedasticity (FIGARCH). Recently, to capture the long-memory property in count data of high frequency, Quoreshi (2014) developed an integer-valued ARFIMA (INARFIMA) model.

Volatility is considered as a key element in estimations under capital asset pricing, portfolio and risk management, derivatives pricing and such other models used in financial market analysis. The introduction of the long memory in financial markets analysis (Greene and Fielitz (1977) and Aydogan and Booth (1988)), spawned a large number of studies investigating financial assets' return and volatility. Moreover, volatility behavior during turmoil has also come to the attention of researchers. The studies intend to confirm whether the long memory property is common in financial markets along with measuring the property and its implication for investments. A number of those studies (Hiemstra and Jones 1997; Willinger et al. 1999; Sadique and Silvapulle 2001; Cavalcante and Assaf 2002; Limam 2003) examine long memory in returns and volatility to produce mixed results. In another study applying ARFIMA-FIGARCH and GPH (Geweke and Porter-Hudak 1983), the fractal structure provides no support for long-memory though GPH appears significant in limited cases (Berg and Lyhagen 1998). Kang and Yoon (2007) however, estimate return and volatility using an ARFIMA-FIGARCH joint model. He demonstrates that the model is significantly stronger compared to each model individually. Further, the large amount of research lends itself to evidence of long memory in return volatilities across markets and time (Oomen 2001; Lee et al. 2001; Sourial 2002; Koopman and HolUspensky 2002; Degiannakis 2004; Broto and Ruiz 2004; Bellalah et al. 2005; Nielsen

2007; Christensen et al. 2007; Chan and Feng 2008; Louzis et al. 2010; Conrad et al. 2011. Parametric and non-parametric tests by Breidt et al. (1998) also generate evidence in support of long-memory in volatility proxies. Wright (2001) also finds strong evidence for long memory by using a semi-parametric method on proxy measures, squared, log-squared and absolute returns. Grau-Carles (2000) applies GPH and ARFIMA models and concludes persistence in volatility of absolute and squared returns to be evident. Similarly Ray and Tsay (2012) find strong evidence in Standard and Poor's 500 index of long memory in volatility. Cajueiro and Tabak (2005) applied the time-varying Hurst exponent to test the long-range dependency of volatility for developed and emerging markets' stock returns. They find strong evidence for the hypothesis (long rage dependence). Powera and Turvey (2010) also find evidence for long-range dependence. Interestingly they apply a different approach; an improved Hurst coefficient estimator and test fourteen energy and agriculture commodities' volatility. Hence, the choice of a long memory model is obvious in estimating squared index return series.

Quoreshi and Mollah (2019) develop a long memory model incorporating conditional heteroscedasticity properties and subsequently apply the model for the squared returns of stock indices using data from BRICS countries, UK and US markets. The model is called fractionally integrated moving average conditional heteroskedasticity (FIMACH). The model, designed for non-integer data, follows Quoreshi (2014). One important way the FIMACH model differs from the ARFIMA model class is that it can study the level series for the heteroskedasticity property. The ARFIMA-FIGARCH class, in comparison, studies the same property on the fractionally differenced series applying Fourier transformation. The FIMACH model can measure the response time to news or rumors, and captures information spread across market system. The model is specified in terms of first and second order moments conditioned on historical observations. FIMACH performs better in reducing serial correlations than ARIFIMA-FIGARCH models for application of squared index return. Hence, in the present study, we apply the FIMACH model to make use of the advantages it provides to investigate contagion of return volatilities in equity markets. We employ the definition of contagion as a significant increase in cross-market correlations after the shock (Forbes and Rigobon 2002). If the increase in cross-market correlations is not significant we call the situation interdependence as the author defined.

In this paper we find evidence for contagion in the volatilities of stock index returns. Further analyses show that evidence for contagion in the volatility of stock index returns for a number of countries increases when we employ a predicted series of squared index returns. We also study the explained and unexplained behaviors of contagion.

The paper is organized as follows. The data descriptive and correlation analysis are given in Section 2. The use of FIMACH model for measuring contagion is discussed in the next section. The following section discusses the estimation procedures. Results and analyses of the results are presented in Section 5. The final section comprises the concluding remarks of the study.

## 2. Data Descriptive and Correlation Analysis

Stock data is collected from various sources, including stock exchanges, Yahoo! Finance, Investing.com, and Stooq. The dataset contains squared stock index returns generated from closing prices of 35 stocks markets from 32 countries, including the USA, UK, majority of Euro Zone, and BRICS countries. We use the terms stock index or index to refer squared index returns. The time period covered is 2 January 2003 to 19 February 2019. The periods in the analysis of correlations are specifically defined as follows: (i) Pre-Global Financial Crisis (Pre-GFC): 18 March 2005 to 8 August 2007, (ii) Global Financial Crisis (GFC): 9 August 2007 to 31 December 2009, (iii) Post-GFC/Pre-Euro Zone Crisis (Pre-EZC): 2 January 2010 to 1 May 2010, (iv) Euro Zone Crisis (EZC): 2 May 2010 to 16 February 2012, and (v) Post-Euro Zone Crisis (Post-EZC): 17 February 2012 to 19 February 2019. All stocks are merged into one dataset with synchronized number of trading days. A non-trading weekday within a country is replaced with the closing price from the previous trading day. Total observations in the main dataset stand at 4227. The squared stock index return is used as a volatility measure of

stock indices (Quoreshi and Mollah 2019). We use the terms level series or volatility of stock indexes to refer the squared stock index return series. Mollah et al. (2016) show that during both crises contagion spread from the USA to other markets. Hence, the cross-correlation coefficients of the volatility measure between the three US stock index return and the rest of the stock markets are shown in Table 1. It represents also the same correlation coefficients between three major Euro Zone countries (Germany, France and Italy) and the rest of the stock used in order to see if the volatility of stock index return of these countries has an impact on the volatility of the stock index return of the other countries. It shows the correlations between the squared stock index return of the USA with the other 32 countries across the world. The correlation coefficients for DJI and Germany increase from 0.381 during the Pre-GFC period to 0.674 during the GFC period, and thereafter decrease to 0.582 during post-GFC. A similar pattern is observed for the other two indexes (NASDAQ and S&P 500) with Germany. This indicates that there may be a spread of contagion from the volatility of stock index returns of the USA to Germany. A summary of possible contagion for EZC is presented in Table 2.

**Table 1.** Correlation coefficients for squared stock index return for Pre-GFC, GFC and Post-GFC.

| Index [1] | DJI | | | NASDAQ | | | S&P 500 | | |
|---|---|---|---|---|---|---|---|---|---|
| | Pre-Crisis | During Crisis | Post-Crisis | Pre-Crisis | During Crisis | Post-Crisis | Pre-Crisis | During Crisis | Post-Crisis |
| AT | 0.161 | 0.473 | 0.267 | 0.097 | 0.504 | 0.226 | 0.163 | 0.491 | 0.298 |
| BE | 0.355 | 0.530 | 0.526 | 0.257 | 0.557 | 0.474 | 0.344 | 0.546 | 0.601 |
| DE | 0.381 | 0.674 | 0.582 | 0.409 | 0.638 | 0.565 | 0.375 | 0.652 | 0.649 |
| ES | 0.355 | 0.445 | 0.672 | 0.326 | 0.435 | 0.625 | 0.349 | 0.448 | 0.751 |
| FI | 0.321 | 0.453 | 0.385 | 0.288 | 0.492 | 0.427 | 0.327 | 0.480 | 0.485 |
| FR | 0.406 | 0.481 | 0.488 | 0.365 | 0.508 | 0.460 | 0.400 | 0.495 | 0.551 |
| GR | 0.216 | 0.244 | 0.195 | 0.187 | 0.246 | 0.219 | 0.229 | 0.253 | 0.226 |
| IR | 0.229 | 0.286 | 0.372 | 0.176 | 0.355 | 0.331 | 0.236 | 0.340 | 0.416 |
| IT | 0.361 | 0.500 | 0.517 | 0.313 | 0.522 | 0.518 | 0.355 | 0.502 | 0.604 |
| MT | −0.010 | 0.027 | 0.022 | −0.030 | 0.031 | 0.040 | −0.023 | 0.030 | 0.018 |
| NL | 0.336 | 0.516 | 0.525 | 0.349 | 0.562 | 0.534 | 0.330 | 0.552 | 0.605 |
| PT | 0.183 | 0.378 | 0.564 | 0.121 | 0.387 | 0.559 | 0.193 | 0.370 | 0.659 |
| UK | 0.384 | 0.436 | 0.648 | 0.298 | 0.462 | 0.558 | 0.390 | 0.460 | 0.695 |
| BR | 0.450 | 0.777 | 0.774 | 0.419 | 0.746 | 0.644 | 0.481 | 0.775 | 0.806 |
| CN | 0.216 | 0.037 | −0.041 | 0.122 | 0.051 | −0.062 | 0.213 | 0.038 | −0.042 |
| IN | 0.058 | 0.168 | 0.169 | 0.087 | 0.155 | 0.055 | 0.068 | 0.158 | 0.134 |
| RU | 0.039 | 0.176 | 0.177 | 0.049 | 0.180 | 0.101 | 0.046 | 0.192 | 0.160 |
| ZA | 0.100 | 0.280 | 0.118 | 0.090 | 0.277 | 0.021 | 0.121 | 0.270 | 0.127 |
| BG | 0.028 | 0.181 | −0.075 | 0.015 | 0.167 | −0.059 | 0.025 | 0.168 | −0.051 |
| CZ | 0.192 | 0.246 | 0.369 | 0.178 | 0.251 | 0.403 | 0.206 | 0.247 | 0.389 |
| HU | 0.090 | 0.377 | 0.377 | 0.048 | 0.339 | 0.343 | 0.083 | 0.381 | 0.413 |
| PL | 0.185 | 0.227 | 0.387 | 0.177 | 0.222 | 0.400 | 0.190 | 0.251 | 0.421 |
| RO | 0.007 | 0.208 | −0.003 | 0.000 | 0.212 | −0.015 | 0.010 | 0.204 | −0.025 |
| EE | 0.136 | 0.129 | 0.136 | 0.127 | 0.094 | 0.142 | 0.159 | 0.126 | 0.089 |
| HR | 0.002 | 0.546 | 0.052 | −0.031 | 0.525 | 0.087 | −0.007 | 0.533 | 0.089 |
| LT | 0.037 | 0.303 | 0.035 | 0.078 | 0.258 | 0.083 | 0.045 | 0.292 | 0.050 |
| LV | 0.039 | 0.242 | 0.022 | 0.015 | 0.241 | 0.032 | 0.045 | 0.250 | 0.026 |
| DK | 0.189 | 0.396 | 0.227 | 0.151 | 0.402 | 0.191 | 0.189 | 0.402 | 0.206 |
| NO | 0.151 | 0.422 | 0.487 | 0.133 | 0.474 | 0.492 | 0.177 | 0.467 | 0.512 |
| SE | 0.314 | 0.428 | 0.399 | 0.269 | 0.476 | 0.432 | 0.308 | 0.461 | 0.468 |
| HK | 0.079 | 0.415 | 0.077 | 0.091 | 0.338 | 0.082 | 0.095 | 0.374 | 0.085 |
| JP | −0.011 | 0.165 | 0.013 | 0.016 | 0.153 | 0.070 | −0.008 | 0.143 | 0.013 |

[1] Abbreviation of country code for the index are given in Table A5 in Appendix A.

**Table 2.** Correlation coefficients for squared stock index return for Pre-EZC, EZC and Post-EZC.

| Index | Germany | | | France | | | Italy | | |
|---|---|---|---|---|---|---|---|---|---|
| | Pre-Crisis | During Crisis | Post-Crisis | Pre-Crisis | During Crisis | Post-Crisis | Pre-Crisis | During Crisis | Post-Crisis |
| AT | 0.478 | 0.740 | 0.727 | 0.589 | 0.839 | 0.751 | 0.636 | 0.753 | 0.718 |
| BE | 0.718 | 0.737 | 0.869 | 0.849 | 0.929 | 0.901 | 0.790 | 0.857 | 0.799 |
| ES | 0.669 | 0.540 | 0.712 | 0.630 | 0.845 | 0.833 | 0.783 | 0.876 | 0.907 |
| FI | 0.707 | 0.830 | 0.546 | 0.736 | 0.879 | 0.496 | 0.626 | 0.799 | 0.364 |
| GR | 0.336 | 0.300 | 0.348 | 0.426 | 0.348 | 0.376 | 0.305 | 0.331 | 0.400 |
| IR | 0.520 | 0.642 | 0.551 | 0.617 | 0.795 | 0.541 | 0.483 | 0.733 | 0.545 |
| MT | 0.056 | -0.021 | −0.009 | −0.014 | −0.009 | −0.005 | 0.021 | −0.003 | 0.002 |
| NL | 0.791 | 0.821 | 0.881 | 0.816 | 0.949 | 0.900 | 0.795 | 0.848 | 0.720 |
| PT | 0.648 | 0.508 | 0.623 | 0.670 | 0.802 | 0.677 | 0.664 | 0.816 | 0.653 |
| UK | 0.844 | 0.717 | 0.680 | 0.868 | 0.733 | 0.700 | 0.748 | 0.659 | 0.499 |
| BR | 0.661 | 0.458 | 0.129 | 0.615 | 0.431 | 0.137 | 0.674 | 0.340 | 0.127 |
| CN | 0.039 | 0.082 | 0.233 | 0.057 | 0.056 | 0.200 | 0.066 | 0.024 | 0.131 |
| IN | 0.149 | 0.302 | 0.295 | 0.309 | 0.346 | 0.317 | 0.292 | 0.300 | 0.217 |
| RU | 0.148 | 0.512 | 0.202 | 0.197 | 0.399 | 0.162 | 0.164 | 0.289 | 0.116 |
| ZA | 0.149 | 0.123 | 0.157 | 0.057 | 0.103 | 0.168 | 0.055 | 0.102 | 0.170 |
| BG | 0.023 | 0.053 | 0.052 | 0.089 | 0.007 | 0.064 | 0.145 | −0.006 | 0.055 |
| CZ | 0.292 | 0.530 | 0.530 | 0.434 | 0.636 | 0.523 | 0.487 | 0.574 | 0.495 |
| HU | 0.362 | 0.515 | 0.391 | 0.348 | 0.732 | 0.399 | 0.420 | 0.712 | 0.376 |
| PL | 0.394 | 0.723 | 0.468 | 0.545 | 0.672 | 0.462 | 0.542 | 0.612 | 0.384 |
| RO | 0.050 | 0.299 | 0.072 | 0.122 | 0.474 | 0.075 | 0.076 | 0.416 | 0.083 |
| EE | 0.383 | 0.438 | 0.245 | 0.335 | 0.413 | 0.254 | 0.178 | 0.311 | 0.173 |
| HR | 0.131 | 0.100 | 0.184 | 0.144 | 0.088 | 0.202 | 0.298 | 0.044 | 0.170 |
| LT | 0.170 | 0.151 | 0.190 | 0.423 | 0.155 | 0.170 | 0.346 | 0.098 | 0.105 |
| LV | 0.034 | 0.143 | 0.054 | 0.077 | 0.198 | 0.039 | 0.028 | 0.180 | 0.012 |
| DK | 0.286 | 0.659 | 0.409 | 0.261 | 0.772 | 0.379 | 0.173 | 0.674 | 0.300 |
| NO | 0.588 | 0.759 | 0.522 | 0.450 | 0.766 | 0.544 | 0.569 | 0.689 | 0.406 |
| SE | 0.661 | 0.879 | 0.547 | 0.540 | 0.837 | 0.470 | 0.493 | 0.742 | 0.343 |
| HK | 0.238 | 0.284 | 0.267 | 0.331 | 0.261 | 0.294 | 0.233 | 0.174 | 0.222 |
| JP | 0.037 | 0.112 | 0.312 | 0.224 | 0.069 | 0.351 | 0.186 | 0.020 | 0.362 |
| DJI | 0.582 | 0.622 | 0.378 | 0.488 | 0.585 | 0.377 | 0.517 | 0.547 | 0.296 |
| NASDAQ | 0.565 | 0.586 | 0.365 | 0.460 | 0.562 | 0.357 | 0.518 | 0.506 | 0.292 |
| S&P500 | 0.649 | 0.600 | 0.411 | 0.551 | 0.565 | 0.416 | 0.604 | 0.517 | 0.332 |

## 3. FIMACH Model for Correlation

Let $p_t$ is price for an index at time $t$. Hence, $r_t = p_t - p_{t-1}$ can be defined as stock index return. If the expected value of $r_t$ is zero, we may consider $r_t^2$ as variance at time point $t$. Assume that stock index return volatility $r_t^2$ has an autocorrelation function which decays very slowly. Note that the square root of the variance, i.e., standard deviation, is extensively used as a measure of volatility. For simplicity, we assume that $x_t$ represents $r_t^2$, stock index return volatility. Assume that $x_t$ is a time series which takes only real values over discrete time. The ARFIMA (p, d, q) model for the series is

$$\alpha(L)(1 - L)^d x_t = \beta(L)u_t. \tag{1}$$

(Granger and Joyeux 1980; Hosking 1981). The ARFIMA (0, d, 0) of the series $x_t$ is then

$$x_t = u_t + d_1 u_{t-1} + d_2 u_{t-2} + d_3 u_{t-3} \ldots$$

or

$$x_t = (1 + L)^{-d} u_t. \tag{2}$$

The $x_t$ has long memory which implies that the autocorrelation function of the series decay slowly. The $u_t$ has zero-mean and assumed to be serially uncorrelated. The parameters $d_j = \Gamma(j + d) / [\Gamma(j + 1)\Gamma(d)]$ where $j = 0, 1, 2, \ldots$ with $d_0 = 1$. Granger and Joyeux (1980) propose that the $d_j$ may be approximated by $Aj^{-d}$, for $j \geq 1$. Quoreshi and Mollah (2019) assume that the $u_t$ is an independent and identically distributed (i.i.d.) sequence of random variables. The unconditional

mean for $u_t$ is $E(u) = \lambda$ and unconditional variance is $V(\alpha u) = \alpha^2 \varnothing^2$ where $V(u) = E(u)^2 - \lambda^2 = \varnothing^2$. The corresponding conditional moments are $E(u|u) = u$ and $V(\alpha u|u) = \alpha^2 V(u|u)$ where $V(u|u) = u^2 - 2\lambda u + \lambda^2$. Under these assumptions, the conditional mean and variance for FIMACH are

$$E(x_t|Y_{t-1}) = E_{t-1} = \lambda + \sum_{i=1}^{m} d_i u_{t-i} \tag{3}$$

and

$$V(x_t|Y_{t-1}) = V_{t-1} = \varnothing^2 + \sum_{i=1}^{m} d_i^2 \left( u_{t-i}^2 - 2\lambda u_{t-i} + \lambda^2 \right) \tag{4}$$

The $Y_{t-1}$ is the information set at time $t-1$ and $m = \infty$. Quoreshi and Mollah (2019) claim that the model is different from the model introduced by Granger and Joyeux (1980) and Hosking (1981) since the conditional mean and variance are different. Note that these two moments vary with $u_{t-j}$. Hence, there is a conditional heteroskedasticity property (Brännäs and Hall 2001). For $\{x_t\}$ to be a stationary sequence, it is sufficient that $\sum_{j=1}^{\infty} d_j < \infty$. According to the authors, the FIMACH models conditional expected value for $x_t$ while the renowned FIGARCH models long memory property of the variance of the error term $u_t$. The FIMACH model is developed to capture long memory property in squared return for stock index data. In this paper, in line with Quoreshi and Mollah (2019), we use the squared returns of the stock index as a volatility measure. Moreover, Quoreshi and Mollah (2019) show that FIMACH performs better than FIGARCH and GARCH models in terms of removing serial correlations. In this paper, we intend to study the contagion effect between squared return of stock indexes. Hence, the FIMACH model is called for.

Assuming $E(u_t u_t|Y_{t-1}) = u_t^2$ and $E\left(u_t u_{t-j}|Y_{t-1}\right) = 0$, the autocorrelation functions at lag $k$ for FIMACH is

$$\rho_{k|t-1} = \frac{\sum_{j=0}^{\infty} d_j d_{k+j} u_{t-j-k}^2}{V\left(\sigma_t^2|Y_{t-1}\right)} \tag{5}$$

where $k = -j, j$ and $j = 1, 2, \ldots, \infty$ with $d_0 = 1$. This autocorrelation function varies with $u_{t-j}$ which catches heteroscedasticity property in autocorrelation function. Ding et al. (1993) illustrated the heteroscedasticity in autocorrelation function for absolute return of stock. To be noted, for explaining the autocorrelation, the authors assume a smooth function.

In our study we use 35 squared stock indexes return series, hence we need to index the model in Equation (2) as

$$x_{jt} = u_{jt} + d_{j1} u_{jt-1} + d_{j2} u_{jt-2} + d_{j3} u_{jt-3} \ldots \tag{6}$$

where $j = 1, 2, \ldots, 35$ representing different squared stock indexes for our study. The properties of all the parameters are the same as for Equation (2) and the moment conditions are the same as in Equations (3)–(5). In this paper, we investigate the contagion between the predicted values of the stock indices and between the residuals. The predicted values for squared stock index $j$ are

$$\hat{x}_{jt} = \hat{\lambda}_j + \sum_{i=1}^{m} \hat{d}_{ji} u_{jt-i} \tag{7}$$

where $\hat{\lambda}_j$ and $\hat{d}_{ji}$ are estimates of the corresponding parameters and hence the corresponding residuals are

$$\hat{e}_{jt} = x_{jt} - \hat{x}_{jt}, \tag{8}$$

This implies that for any stock index series $x_{jt}$,

$$x_{jt} = \hat{x}_{jt} + \hat{e}_{jt}. \tag{9}$$

Here, we see that the stock index series $x_{jt}$ can be decomposed by its predicted series $\hat{x}_{jt}$ and the residuals $\hat{e}_{jt}$. The predicted values are the explained part of $x_{jt}$ that are captured by the model given in Equation (6). The residuals $\hat{e}_{jt}$ are the unexplained part of $x_{jt}$ that are assumed to be i.i.d. with expected values zero. The $u_{jt-i}$ in Equation (7) are shocks related to the stock index return $j$ that capture information relevant to that particular stock, and $\hat{\lambda}_j$ and $\hat{d}_{ji}$ filter the information. The stock market analyst may use the predicted values to predict the future or use this kind of information, e.g., to diversify the portfolios. Hence, it is important to study the contagion for predicted values between the stock index return series and we call this behavior contagion in predicted volatility. This measure can be viewed as contagion in predicted behaviors of volatility in stock markets. It is also important to investigate the existence of contagion between the residuals which we call contagion in volatility residuals. This measure can be viewed as contagion in unpredicted behaviors in stock markets. Hence, the contagion of level series is the result of the combination of predicted and unpredicted behaviors of volatility in stock markets.

As mentioned earlier we employ the definition of contagion as significant increase in cross-market correlations after the shock (Forbes and Rigobon 2002). If the increase in cross-market correlations is not significant we call the situation interdependence according to the definition of the authors. Hence, the null hypothesis ($H_0$) for contagion is that there is no significant difference between the correlations of two stock series volatilities for the period Pre-GFC and GFC. The alternative hypothesis ($H_a$) is that there is significant difference between the correlations of two stock series volatilities for the period Pre-GFC and GFC; hence, there is evidence for contagion. If the difference in correlation coefficient between the GFC and Pre-GFC is significantly different from zero, we may conclude there is contagion due to GFC. Similarly, if the difference in correlation coefficient between the EZC and Pre-EZC is significantly different from zero, we may conclude there is contagion due to the EZC. These can be written as:

$$Cont_{SP} = \frac{\rho_{j,l}^{SP} - \rho_{j,l}^{Pre-SP}}{\sqrt{V\left(\rho_{j,l}^{TP}\right) + V\left(\rho_{j,l}^{Pre-TP}\right)}} \tag{10}$$

where $Cont_{SP}$ stands for contagion for a certain shock period (SP) which refer to GFC or EZC. The $\rho_{j,l}^{SP}$ is the correlation coefficient for the time period of interest and the $\rho_{j,l}^{Pre-SP}$ correlation coefficient for the previous time period of interest. The $V(\cdot)$ represents variance for corresponding correlation coefficient. If $Cont_{SP} > T-statistics$ we reject the null hypothesis in favor of alternative hypothesis and conclude that there is evidence for contagion in volatility between two stock series due to global financial crisis or Euro Zone Crisis. If $Cont_{SP} < T-statistics$, we cannot reject the null hypothesis and conclude that there is no evidence for contagion in volatility between the two stock series due to the global financial crisis or EZC.

## 4. Estimation

The evidence of long memory property in squared index returns has been found in previous studies (e.g., Bhardwaj and Swanson 2006; Quoreshi and Mollah 2019). The ARFIMA (p, d, q) is introduced by Granger and Joyeux (1980) and Hosking (1981) while the FIGARCH (k, d, l), is introduced by Baillie et al. (1996). Fourier transformation of the level series and autocorrelation function are used for estimation of long memory parameter in these models. Quoreshi and Mollah (2019) show that estimating long memory property using conditional mean function outperforms ARFIMA and FIGACH. Hence, we define, in line with Quoreshi and Mollah, the loss function as:

$$e_{jt} = x_{jt} - E_{jt-1} = x_{jt} - \lambda_j - \sum_{i=1}^{m} d_{ji} u_{jt-i} \tag{11}$$

where $j$ is any time series of squared index return for a particular stock market. The $E_{jt-1}$ the coitional mean of squared index return for market $j$. The $E_{jt-1}$ is defined in Equation (3) without the index $j$.

The criteria $S = \sum_{i=m+1}^{T} e_{jt}^2$ is used in the estimator of interest. Here, $m = 70$ is used as long but finite lag length. This is minimized with respect to unknown parameters, i.e., $\psi = (\lambda_j, \text{ and } d')$. The $d'$ is a vector of parameters with elements $d_i$. In estimating, we restrict $\lambda_j = \exp(L)$ to make sure a positive value for $\lambda_j$. This restriction gives better estimation and faster convergence in estimation procedure. The Quasi-Maximum Likelihood (QML) estimator is used as follows:

$$LnL\left(x_{j1}, x_{j2} \ldots, x_{jT} \middle| Y_{t-1}, \lambda, d_i \text{ and } \hat{V}_{jt-1}\right) = -\ln\left(\hat{V}_{jt-1}\right) - \left(\frac{\sum_{t=m+1}^{T} e_{jt}^2}{\hat{V}_{jt-1}}\right). \tag{12}$$

The $\hat{V}_{jt-1}$ is in accordance with the Equation (4) and is estimated at the same time with the other parameters with start value chosen as suggested by Quoreshi and Mollah (2019). After the estimation of parameters, the predicted values $\hat{x}_{jt}$ and the residuals $\hat{e}_{jt}$ are estimated according to Equations (7) and (8), respectively. The correlation coefficients and the T-test for contagions employing the series $x_{jt}$, $\hat{x}_{jt}$ and $\hat{e}_{jt}$ are calculated in accordance with Equation (10).

## 5. Results

The results from the conditional heteroskedasticity long memory model are presented in Table 3. Both $\hat{\lambda}_j$ and $\hat{d}_j$ are significant. Since the absolute values of $\hat{d}_j$ are less than 0.5, we find the evidence of long memory in squared stock returns of all the 35 stock indices that are in line with previous studies (e.g., Granger and Hyung 1999; Bhardwaj and Swanson 2006 and Quoreshi and Mollah 2019). This implies that the volatility of stock index returns today has a persistent impact on future volatility. The higher the absolute value of $\hat{d}_j$ the greater the impact. The squared stock index return of Lithuania has the largest absolute value of $\hat{d}_j$ (0.31696), while DJI has the smallest absolute corresponding coefficient (0.00916). In general, we may conclude that larger indices have a smaller long memory coefficient. This implies that the impact of volatility in larger markets have smaller impacts on future stock index return volatility compared to the smaller stock index return volatility, although the impact is persistent. This conclusion opens for further research whether the impact is due to the size of the stock index or other characteristics, e.g., difference in country specific factors.

For GFC, we use the three US stock indexes returns as sources of contagion while three major stock indexes of Euro Zone countries (Germany, France and Italy) are used for EZC. We estimate cross-correlations between the sources of contagion and rest of the stock indexes. Besides the level series, predicted series and standardized residual series of the sources of contagion, the lag 1 of these series are also used. For the GFC, lag 1 of the series may be more important compared to the level series to take into account the casual effect due to time differences between the US stock index and rest of the indexes. The cross-correlation coefficients of the volatility measure, for both the predicted and the standardized residual series between the three US stock index returns and the rest of the stock markets, are shown in Tables A1 and A2 in the Appendix A, respectively. The corresponding cross-correlation coefficients between three major Euro Zone countries (Germany, France and Italy) and the rest of the stock are presented in Tables A3 and A4 in Appendix A.

**Table 3.** Estimates of conditional heteroskedasticity long memory model.

| Index | L | s.e. | Exp(L) | s.e. | $\hat{d}_j$ | s.e. | AIC | SBIC | LB100 | LB200 | MSE |
|---|---|---|---|---|---|---|---|---|---|---|---|
| AT | −0.424 | 0.156 | 0.654 | 0.104 | 0.231 | 0.039 | 14,846.367 | 15,487.641 | 811.421 | 1096.774 | 31.952 |
| BE | −0.768 | 0.132 | 0.464 | 0.062 | 0.222 | 0.037 | 11,015.732 | 11,466.528 | 577.753 | 813.322 | 13.095 |
| DE | −4.203 | 0.046 | 0.015 | 0.001 | −0.016 | 0.005 | −25,499.071 | −24,927.638 | 4137.004 | 4648.579 | 0.002 |
| ES | −0.078 | 0.135 | 0.925 | 0.126 | 0.167 | 0.037 | 15,153.824 | 15,464.937 | 336.065 | 433.936 | 35.219 |
| FI | −0.275 | 0.078 | 0.760 | 0.059 | 0.163 | 0.019 | 12,591.763 | 13,233.037 | 842.675 | 1114.709 | 18.744 |
| FR | −4.238 | 0.057 | 0.014 | 0.001 | −0.034 | 0.011 | −25,014.868 | −24,665.660 | 3632.913 | 4054.365 | 0.003 |
| GR | 0.708 | 0.097 | 2.031 | 0.198 | 0.129 | 0.022 | 19,676.834 | 19,880.010 | 335.236 | 674.811 | 103.509 |
| IR | −0.424 | 0.133 | 0.654 | 0.088 | 0.210 | 0.032 | 15,085.916 | 15,727.190 | 954.353 | 1116.235 | 33.815 |
| IT | −0.030 | 0.108 | 0.971 | 0.106 | 0.179 | 0.030 | 15,094.311 | 15,443.520 | 417.294 | 572.725 | 34.628 |
| MT | −1.533 | 0.113 | 0.216 | 0.025 | 0.208 | 0.042 | 2088.162 | 2177.051 | 515.788 | 674.560 | 1.628 |
| NL | −0.334 | 0.121 | 0.716 | 0.088 | 0.215 | 0.042 | 13,629.298 | 13,832.474 | 1574.805 | 1778.028 | 24.753 |
| PT | −0.446 | 0.122 | 0.640 | 0.079 | 0.181 | 0.039 | 11,623.896 | 11,865.168 | 443.252 | 542.819 | 15.359 |
| UK | −4.588 | 0.022 | 0.010 | 0.000 | −0.023 | 0.006 | −27,366.146 | −27,023.287 | 6005.333 | 6875.645 | 0.002 |
| BR | −3.819 | 0.062 | 0.022 | 0.001 | −0.046 | 0.018 | −21,922.137 | −21,503.087 | 4285.984 | 4475.253 | 0.005 |
| CN | −3.777 | 0.042 | 0.023 | 0.001 | −0.006 | 0.002 | −22,974.994 | −21,965.463 | 3762.055 | 5773.820 | 0.004 |
| IN | −4.078 | 0.056 | 0.017 | 0.001 | −0.014 | 0.005 | −22,691.364 | −21,999.296 | 2656.125 | 4108.104 | 0.004 |
| RU | −3.609 | 0.094 | 0.027 | 0.003 | −0.105 | 0.035 | −14,064.406 | −13,892.977 | 1838.231 | 2023.306 | 0.035 |
| SA | −0.574 | 0.112 | 0.563 | 0.064 | 0.189 | 0.026 | 10,895.243 | 11,771.440 | 1054.694 | 1282.237 | 12.330 |
| BG | −1.060 | 0.232 | 0.346 | 0.084 | 0.291 | 0.069 | 11,889.036 | 12,225.546 | 1052.669 | 1384.783 | 16.238 |
| CZ | −0.547 | 0.171 | 0.578 | 0.101 | 0.255 | 0.040 | 16,188.014 | 16,473.730 | 824.646 | 1043.702 | 45.066 |
| HU | −0.332 | 0.281 | 0.718 | 0.214 | 0.241 | 0.080 | 15,278.820 | 15,640.727 | 592.124 | 753.756 | 36.139 |
| PL | −0.408 | 0.075 | 0.665 | 0.050 | 0.150 | 0.018 | 10,002.370 | 10,459.516 | 636.430 | 899.656 | 10.298 |
| RO | −0.436 | 0.186 | 0.647 | 0.123 | 0.234 | 0.042 | 16,845.924 | 17,480.848 | 323.973 | 400.719 | 51.303 |
| EE | −0.880 | 0.207 | 0.415 | 0.089 | 0.188 | 0.046 | 11,085.292 | 11,517.041 | 160.416 | 492.554 | 13.331 |
| HR | −1.244 | 0.458 | 0.288 | 0.155 | 0.324 | 0.144 | 13,652.003 | 13,994.862 | 499.664 | 863.191 | 24.629 |
| LT | −1.294 | 0.316 | 0.274 | 0.093 | 0.317 | 0.092 | 12,326.433 | 12,555.006 | 154.575 | 233.264 | 18.153 |
| LV | −0.474 | 0.165 | 0.622 | 0.105 | 0.184 | 0.042 | 13,271.300 | 13,582.413 | 209.323 | 326.993 | 22.561 |
| DK | −0.493 | 0.136 | 0.611 | 0.084 | 0.198 | 0.038 | 11,983.400 | 12,332.609 | 1406.852 | 1570.633 | 16.588 |
| NO | −0.301 | 0.116 | 0.740 | 0.087 | 0.218 | 0.035 | 15,652.587 | 16,293.861 | 1367.057 | 1651.683 | 38.666 |
| SE | −4.159 | 0.042 | 0.016 | 0.001 | −0.009 | 0.003 | −25,819.084 | −25,120.667 | 7020.940 | 8106.152 | 0.002 |
| HK | −4.170 | 0.085 | 0.015 | 0.001 | −0.035 | 0.022 | −22,538.917 | −21,897.643 | 5465.848 | 6822.842 | 0.005 |
| JP | −4.121 | 0.076 | 0.016 | 0.001 | −0.056 | 0.027 | −23,062.748 | −22,821.476 | 3368.417 | 3581.799 | 0.004 |
| DJI | −4.605 | 0.052 | 0.010 | 0.001 | −0.009 | 0.003 | −26,910.683 | −26,218.615 | 9825.122 | 10,604.954 | 0.002 |
| NASDAQ | −4.274 | 0.045 | 0.014 | 0.001 | −0.012 | 0.003 | −25,491.418 | −24,627.921 | 8930.627 | 9680.700 | 0.002 |
| S&P500 | −4.492 | 0.058 | 0.011 | 0.001 | −0.013 | 0.005 | −25,486.999 | −24,750.486 | 10,390.596 | 11,233.966 | 0.002 |

A summary of T-tests (Lag 1 of Level, Predicted and standardized Residual series of the US Stock Indexes) for contagion as defined in Equation (10) which is statistically significant difference in correlation coefficients between Pre-GFC and GFC in Table 4[1]. The corresponding statistics for EZC are presented in Table 5. We find evidence of contagion from lag 1 of level series of DJI to 20 countries including the two major Euro Zone countries for squared index returns. This implies that the cross-correlations between lag level series of DJI and those squared stock index returns increase significantly (T-statistics > 1.96) for the period of GFC. Note that we do not find any contagion during GFC on squared stock index returns of Germany (DE), but interdependence (T-statistics = 0.666, see Table 4). However, the effect of contagion is obvious on DE if you consider the lag predicted series (T-statistics = 8.360, see Table 4). Employing lag predicted series of DJI, we find contagion on all countries except for the index of Ireland (IR). For IR, we find interdependence. A similar result is found using lag predicted series of NASDAQ and S&P500 as sources of contagion. For NASDAQ, we find contagion on all countries except for the indexes of Ireland, China (CN) and Lithuania (LT). For S&P500, we find contagion on all countries except for Ireland and China. The results indicate that the squared index returns are interdependent.

---

[1]　The estimates of cross-correlations with the lag 1 series are not presented here. The results are available upon requests to the authors.

**Table 4.** T-test for Contagion of GFC for lag Level, Predicted and Residual Series and DJI, NASDAQ and S&P500 are sources of contagion.

| Index | DJI | | | NASDAQ | | | S&P500 | | |
|---|---|---|---|---|---|---|---|---|---|
| | Level | Predicted | Residual | Level | Predicted | Residual | Level | Predicted | Residual |
| AT | 3.412 | 11.130 | 0.349 | 1.789 | 8.935 | −1.344 | 2.945 | 10.825 | −0.123 |
| BE | 2.844 | 6.191 | 5.129 | 2.051 | 4.424 | 4.830 | 2.568 | 5.760 | 4.597 |
| DE | 0.666 | 8.360 | 0.409 | 0.060 | 7.053 | −0.216 | 0.581 | 8.263 | 0.292 |
| ES | 4.637 | 7.900 | 3.411 | 2.842 | 5.812 | 1.529 | 3.737 | 6.786 | 2.546 |
| FI | 0.790 | 7.767 | −0.440 | −0.336 | 5.814 | −1.609 | 0.252 | 7.605 | −1.027 |
| FR | 3.558 | 8.251 | 3.287 | 2.196 | 6.528 | 1.860 | 2.768 | 7.778 | 2.459 |
| GR | 2.960 | 6.418 | 2.264 | 1.617 | 3.711 | 1.198 | 2.432 | 5.810 | 1.795 |
| IR | −0.614 | 1.255 | −1.631 | −0.129 | 1.063 | −1.408 | −0.512 | 0.509 | −1.552 |
| IT | 3.829 | 8.456 | 2.338 | 2.856 | 8.394 | 1.234 | 3.180 | 8.033 | 1.702 |
| MT | 1.680 | 2.648 | 1.200 | 1.927 | 2.533 | 1.551 | 1.872 | 3.018 | 1.334 |
| NL | 1.936 | 6.401 | 0.001 | 0.665 | 5.180 | −1.507 | 1.415 | 6.055 | −0.625 |
| PT | 4.328 | 8.350 | 2.505 | 3.446 | 8.280 | 1.501 | 3.648 | 7.013 | 2.003 |
| UK | 4.043 | 8.503 | 3.834 | 2.353 | 6.933 | 2.078 | 2.934 | 7.359 | 2.701 |
| BR | 1.825 | 8.543 | 1.198 | 1.886 | 7.132 | 1.310 | 1.699 | 8.025 | 1.079 |
| CN | 0.886 | 2.331 | 0.871 | 0.308 | 3.217 | 0.270 | 0.647 | 1.103 | 0.630 |
| IN | −3.553 | 4.367 | −3.737 | −3.658 | 0.908 | −3.790 | −3.313 | 4.071 | −3.499 |
| RU | 4.325 | 8.607 | 3.691 | 2.188 | 5.771 | 1.588 | 3.309 | 7.945 | 2.621 |
| ZA | 6.777 | 12.112 | 5.129 | 6.229 | 9.893 | 4.830 | 6.183 | 11.800 | 4.597 |
| BG | 6.129 | 12.055 | 3.447 | 6.520 | 12.806 | 4.027 | 6.472 | 12.119 | 3.935 |
| CZ | 7.611 | 9.774 | 5.994 | 5.197 | 6.948 | 4.003 | 6.511 | 8.700 | 5.135 |
| HU | 6.270 | 11.844 | 3.836 | 3.563 | 8.814 | 1.535 | 5.049 | 10.932 | 2.672 |
| PL | 5.394 | 10.273 | 4.471 | 2.720 | 5.912 | 2.099 | 4.198 | 9.428 | 3.271 |
| RO | 4.344 | 8.816 | 2.263 | 3.476 | 9.402 | 1.363 | 3.761 | 8.785 | 1.699 |
| EE | 1.274 | 6.696 | 0.499 | 0.727 | 5.267 | 0.208 | 0.951 | 5.392 | 0.388 |
| HR | 1.551 | 9.931 | −3.852 | 1.042 | 9.225 | −4.360 | 1.024 | 9.165 | −4.271 |
| LT | 1.367 | 3.107 | −1.686 | 0.459 | 1.635 | −1.907 | 1.021 | 2.742 | −1.801 |
| LV | 3.152 | 8.719 | 1.355 | 3.450 | 9.261 | 1.851 | 3.392 | 9.249 | 1.642 |
| DK | 2.939 | 9.316 | 0.916 | 1.923 | 7.548 | −0.004 | 2.407 | 8.667 | 0.423 |
| NO | 1.997 | 12.240 | −0.913 | 1.868 | 9.959 | −0.848 | 1.859 | 12.106 | −1.084 |
| SE | 2.451 | 11.393 | 2.280 | 1.427 | 8.731 | 1.231 | 1.800 | 11.216 | 1.581 |
| HK | −3.154 | 6.666 | −3.691 | −4.017 | 3.857 | −4.499 | −4.094 | 4.216 | −4.587 |
| JP | 6.870 | 11.744 | 6.383 | 5.522 | 7.537 | 5.224 | 6.888 | 11.005 | 6.449 |

For lag residual series of DJI, we find contagion on index of 16 countries that may be compared to 31 stock indexes for lag predicted series. This may be interpreted that the 16 of 31 stock markets react based on explained and unexplained information, while rest of the 15 markets react based only on explained information from the source index.

During the EZC, we find no evidence of contagion from lag level series of squared stock index returns of Germany (DE) to the 32 stock indices, including the DJI, NASDAQ and S&P500 (See Table 5). This implies that the cross-correlations between DE and those indexes do not increase significantly (T-statistics < 2) for the period of EZC. However, employing a lag predicted series of DE, we find evidence for contagion on nine of the stock indexes inclusive four Nordic countries (FI, DK, NO & SE), Russia (RU) and South Africa (ZA). Notably, no effect of contagion is observed from DE to the US stock indexes. However, there is evidence for contagion from lag predicted index series of France (FR) to the three US stock indexes while the lag predicted series of Italy (IT) have a contagion impact on only NASDAQ among the three US stock indices. Note also that we observe negative significant T-statistics which imply that the correlations between DE and the stock index of those countries (CN, LT and JP) decrease significantly or go to opposite direction. This may indicate that there is a change in stock market trading behavior for these countries in relation to DE. It is also important to note that post-GFC correlations decrease generally with few exceptions, while post-EZC correlations have rather ambiguous behaviors. What are the impacts of contagions afterwards? This is an open question that need to be addressed in further research.

**Table 5.** T-test for contagion of EZC for lag Level, Predicted and Residual Series and Stock Index of Germany. France and Italy are sources of contagion.

| Index | Germany (DE) | | | France (FR) | | | Italy (IT) | | |
|---|---|---|---|---|---|---|---|---|---|
| | Level | Predicted | Residual | Level | Predicted | Residual | Level | Predicted | Residual |
| AT | −0.625 | 1.875 | −1.662 | −1.337 | 2.121 | −2.265 | −1.876 | 1.530 | −3.104 |
| BE | −1.190 | 0.720 | 0.471 | −1.227 | 1.351 | 0.779 | −1.928 | 1.008 | 0.112 |
| ES | 0.599 | −0.241 | 0.634 | 0.815 | 1.576 | 0.530 | −0.117 | 0.648 | −0.481 |
| FI | 1.417 | 4.097 | 0.677 | 1.013 | 3.684 | 0.395 | 0.532 | 3.218 | −0.271 |
| GR | −0.722 | −0.355 | −0.868 | −0.675 | −0.876 | −0.650 | −0.986 | −0.690 | −0.650 |
| IR | 0.207 | 1.433 | −0.380 | 0.171 | 2.786 | −0.507 | −0.488 | 2.540 | −1.487 |
| MT | 1.198 | −0.241 | 1.428 | 1.481 | 1.653 | 1.388 | 0.388 | 1.704 | 0.093 |
| NL | 0.389 | 2.310 | −0.283 | 0.480 | 2.738 | −0.198 | −0.361 | 1.657 | −1.244 |
| PT | 0.872 | 0.059 | 1.078 | 1.156 | 1.377 | 1.116 | 0.169 | 0.991 | 0.018 |
| UK | −2.109 | −1.230 | −2.178 | −2.297 | −0.289 | −2.413 | −1.210 | 0.212 | −1.640 |
| BR | 1.220 | 1.034 | 1.164 | 1.017 | 1.312 | 1.004 | 0.046 | 0.675 | −0.155 |
| CN | −1.277 | −3.137 | −1.291 | −0.051 | 0.477 | −0.126 | −0.752 | −0.064 | −1.263 |
| IN | −3.005 | −1.152 | −3.069 | −2.224 | −1.126 | −2.293 | −3.371 | −1.645 | −3.765 |
| RU | −0.059 | 2.297 | −0.553 | −0.366 | 2.588 | −0.785 | −0.201 | 2.264 | −0.894 |
| ZA | 0.765 | 2.307 | 0.471 | 1.028 | 2.684 | 0.779 | 0.386 | 1.391 | 0.112 |
| BG | 0.320 | −0.914 | 0.200 | 0.318 | −0.153 | 0.488 | 0.337 | 0.272 | 0.519 |
| CZ | −0.841 | 1.138 | −1.864 | −0.266 | 1.888 | −1.076 | −1.132 | 0.891 | −2.085 |
| HU | 0.286 | 3.296 | −0.444 | 0.469 | 3.801 | −0.663 | −0.534 | 2.780 | −1.901 |
| PL | −1.481 | 2.466 | −2.510 | −1.188 | 1.323 | −1.720 | −2.538 | −0.214 | −3.270 |
| RO | −0.041 | 0.648 | −0.713 | 0.829 | 3.047 | −0.129 | 0.894 | 3.243 | −0.299 |
| EE | −0.654 | 0.292 | −1.142 | −0.553 | 1.882 | −1.107 | −1.891 | 1.289 | −3.041 |
| HR | −0.291 | −1.468 | 0.040 | −0.284 | −0.336 | 0.118 | 0.036 | −0.092 | 0.737 |
| LT | −2.977 | −3.268 | −2.978 | −3.080 | −3.158 | −2.327 | −4.177 | −4.135 | −3.667 |
| LV | 0.492 | 0.078 | 0.340 | 0.165 | 1.098 | −0.136 | −0.459 | 0.425 | −0.869 |
| DK | 0.541 | 3.120 | −0.596 | 0.078 | 3.672 | −1.234 | −0.238 | 3.573 | −1.700 |
| NO | 1.704 | 3.072 | 0.742 | 1.014 | 3.240 | 0.022 | 0.569 | 2.485 | −0.271 |
| SE | 1.688 | 3.765 | 1.593 | 1.354 | 3.508 | 1.240 | 0.302 | 2.765 | −0.089 |
| HK | 0.201 | 2.591 | 0.048 | 0.131 | 1.613 | 0.029 | −1.196 | 0.724 | −1.747 |
| JP | −4.464 | −5.795 | −4.448 | −3.339 | −4.268 | −3.260 | −3.077 | −4.370 | −2.946 |
| DJI | 0.842 | 1.570 | 0.788 | 0.679 | 2.372 | 0.590 | 0.201 | 1.789 | −0.060 |
| NASDAQ | 1.596 | 1.409 | 1.558 | 1.371 | 2.611 | 1.306 | 0.775 | 1.984 | 0.578 |
| S&P500 | 1.270 | 1.442 | 1.219 | 1.074 | 2.186 | 0.989 | 0.632 | 1.510 | 0.350 |

The empirical results show that there is contagion in volatility of stock index returns for predicted and unpredicted behaviors. This may imply that actors in the stock markets have reacted based on information that are of interest for a particular stock market. They may also have reacted based on just rumors or trend or nervousness. It is clear from the figures and the tables that the reactions are different in different countries. In summary, we conclude that there is evidence for contagion and interdependence of squared stock index returns during the GFC and EZC that is in line with previous studies (e.g., Wang et al. 2017; Gamba-Santamaria et al. 2017; Jiang et al. 2017; Bonga-Bonga 2018). Wang et al. (2017) find evidence for contagion during the GFC on G7 countries (except for Japan), Russia and India where US is used as sources of contagion and no contagion is found on Brazil, China, and Japan from the same source. Note that we find evidence for contagion on all the predicted indices of BRICS countries and Japan, where the DJI is the source of contagion. But the results are mixed when employing NADAQ and S&P500 as sources of contagion. Jiang et al. (2017) that the correlation of stock markets between the US, Britain, Germany, Japan and Hong Kong increases markedly after the crisis, while it exhibits a reverse trend with the Chinese stock market. In this study, we find evidence for contagion for all these countries for lag predicted series of DJI as sources of contagion. Note that we study contagion for squared stock index returns, while the previous studies consider stock returns or stock index returns.

## 6. Concluding Remarks

In summary, we find evidence for contagion during the GFC using lag level series of DJI, Nasdaq and S&P 500 as sources for contagion. Similar results are found for the EZC where stock indices of Germany, France and Italy are used as sources of contagion. The intensity (magnitude of cross-correlations) of contagion varies depending on the sources of contagion. We observe that the effects of GFC are different on different stock indexes and it varies depending on sources of contagion. We find also evidence for contagion using lag predicted series and standardized residuals series. These series decompose the total effect which is visible in the level series. The evidence from predicted series illuminates the explained behavior of the stock indices while the residual series capture the unexplained behavior. For the lag predicted series of DJI, we see that the cross-country correlations increase significantly for 31 of 32 observed stock indexes during GFC. Similar results are observed for NASDAQ and S&P500, although the effects are visible on fewer squared stock indexes. For lag residual series of DJI, we find contagion on the indices of 16 countries that may be compared to 31 stock indexes for the lag predicted series. During the EZC, we find no evidence of contagion from the lag level series of squared stock index returns of Germany (DE) to the 32 stock index inclusive DJI, NASDAQ and S&P500. However, employing lag predicted series of DE, we find evidence for contagion on nine of the stock indices, including four Nordic countries (FI, DK, NO & SE), Russia (RU) and South Africa (ZA). Hence, it is important to decompose the explained and unexplained behavior in order to capture the effect of contagion. We also observe that post-GFC and post-EZC correlations do not decrease univocally, which requires further attention to investigate.

**Author Contributions:** R.U. contributed to literature review, estimation and data Analysis. V.J. contributed to estimation, data analysis and commenting on overall paper. A.M.M.S.Q. designed, supervised and wrote the paper.

**Funding:** This research received no external funding.

**Conflicts of Interest:** The authors declare no conflict of interest.

## Appendix A

**Table A1.** Cross correlations coefficients between lag predicted series during Pre-GFC. GFC and post-GFC.

| Index | DJI | | | NASDAQ | | | S&P 500 | | |
|---|---|---|---|---|---|---|---|---|---|
| | Pre-Crisis | During Crisis | Post-Crisis | Pre-Crisis | During Crisis | Post-Crisis | Pre-Crisis | During Crisis | Post-Crisis |
| AT | −0.091 | 0.842 | 0.544 | −0.138 | 0.824 | 0.558 | −0.108 | 0.840 | 0.599 |
| BE | 0.081 | 0.757 | 0.512 | 0.048 | 0.735 | 0.498 | 0.074 | 0.751 | 0.616 |
| DE | 0.727 | 0.907 | 0.731 | 0.723 | 0.879 | 0.756 | 0.724 | 0.895 | 0.790 |
| ES | 0.051 | 0.789 | 0.644 | 0.046 | 0.740 | 0.639 | 0.052 | 0.769 | 0.748 |
| FI | 0.147 | 0.795 | 0.302 | 0.145 | 0.799 | 0.339 | 0.152 | 0.803 | 0.423 |
| FR | 0.222 | 0.856 | 0.611 | 0.269 | 0.832 | 0.601 | 0.241 | 0.848 | 0.693 |
| GR | 0.117 | 0.611 | 0.265 | 0.097 | 0.559 | 0.287 | 0.118 | 0.588 | 0.328 |
| IR | −0.009 | 0.588 | 0.251 | −0.053 | 0.593 | 0.241 | −0.020 | 0.601 | 0.322 |
| IT | 0.055 | 0.836 | 0.601 | 0.065 | 0.820 | 0.609 | 0.061 | 0.826 | 0.705 |
| MT | −0.090 | 0.050 | 0.098 | −0.120 | 0.059 | 0.126 | −0.099 | 0.060 | 0.071 |
| NL | 0.174 | 0.804 | 0.504 | 0.257 | 0.785 | 0.519 | 0.203 | 0.802 | 0.604 |
| PT | −0.003 | 0.660 | 0.414 | −0.024 | 0.617 | 0.428 | −0.005 | 0.638 | 0.547 |
| UK | 0.096 | 0.852 | 0.727 | 0.095 | 0.822 | 0.710 | 0.100 | 0.844 | 0.783 |
| BR | −0.023 | 0.879 | 0.836 | −0.038 | 0.835 | 0.768 | −0.027 | 0.864 | 0.875 |
| CN | 0.927 | 0.312 | 0.236 | 0.971 | 0.293 | 0.288 | 0.946 | 0.299 | 0.260 |
| IN | 0.819 | 0.607 | 0.676 | 0.762 | 0.578 | 0.630 | 0.810 | 0.589 | 0.645 |
| RU | −0.064 | 0.556 | 0.301 | 0.004 | 0.527 | 0.289 | −0.045 | 0.552 | 0.321 |
| ZA | 0.076 | 0.806 | 0.378 | 0.071 | 0.798 | 0.326 | 0.079 | 0.807 | 0.440 |
| BG | 0.148 | 0.616 | 0.253 | 0.133 | 0.593 | 0.234 | 0.152 | 0.609 | 0.245 |

**Table A1.** *Cont.*

| Index | DJI | | | NASDAQ | | | S&P 500 | | |
|---|---|---|---|---|---|---|---|---|---|
| | Pre-Crisis | During Crisis | Post-Crisis | Pre-Crisis | During Crisis | Post-Crisis | Pre-Crisis | During Crisis | Post-Crisis |
| CZ | 0.021 | 0.703 | 0.567 | 0.016 | 0.650 | 0.572 | 0.025 | 0.676 | 0.613 |
| HU | −0.040 | 0.748 | 0.263 | −0.060 | 0.695 | 0.258 | −0.044 | 0.726 | 0.336 |
| PL | 0.224 | 0.711 | 0.514 | 0.281 | 0.681 | 0.489 | 0.246 | 0.706 | 0.581 |
| RO | −0.110 | 0.612 | 0.169 | −0.158 | 0.596 | 0.213 | −0.125 | 0.600 | 0.163 |
| EE | 0.093 | 0.503 | 0.423 | 0.134 | 0.466 | 0.497 | 0.117 | 0.495 | 0.402 |
| HR | −0.018 | 0.699 | 0.207 | −0.036 | 0.669 | 0.266 | −0.020 | 0.683 | 0.254 |
| LT | 0.229 | 0.442 | 0.479 | 0.290 | 0.410 | 0.542 | 0.263 | 0.431 | 0.524 |
| LV | 0.063 | 0.503 | −0.066 | 0.038 | 0.488 | −0.016 | 0.055 | 0.504 | −0.025 |
| DK | 0.182 | 0.777 | 0.113 | 0.202 | 0.740 | 0.132 | 0.191 | 0.762 | 0.155 |
| NO | −0.046 | 0.846 | 0.602 | −0.076 | 0.841 | 0.573 | −0.053 | 0.852 | 0.613 |
| SE | 0.977 | 0.868 | 0.520 | 0.920 | 0.873 | 0.542 | 0.971 | 0.876 | 0.590 |
| HK | 0.917 | 0.733 | 0.540 | 0.815 | 0.668 | 0.515 | 0.884 | 0.701 | 0.534 |
| JP | 0.100 | 0.769 | 0.417 | 0.232 | 0.711 | 0.415 | 0.132 | 0.738 | 0.424 |

**Table A2.** Cross correlations coefficients between lag residual series during Pre-GFC. GFC and Post-GFC.

| Index | DJI | | | NASDAQ | | | S&P 500 | | |
|---|---|---|---|---|---|---|---|---|---|
| | Pre-Crisis | During Crisis | Post-Crisis | Pre-Crisis | During Crisis | Post-Crisis | Pre-Crisis | During Crisis | Post-Crisis |
| AT | 0.147 | 0.292 | 0.226 | 0.095 | 0.338 | 0.195 | 0.146 | 0.319 | 0.266 |
| BE | 0.328 | 0.394 | 0.524 | 0.229 | 0.434 | 0.470 | 0.305 | 0.415 | 0.612 |
| DE | 0.357 | 0.666 | 0.578 | 0.344 | 0.631 | 0.562 | 0.323 | 0.644 | 0.647 |
| ES | 0.335 | 0.337 | 0.648 | 0.305 | 0.332 | 0.601 | 0.316 | 0.346 | 0.735 |
| FI | 0.302 | 0.367 | 0.380 | 0.259 | 0.407 | 0.430 | 0.290 | 0.395 | 0.483 |
| FR | 0.389 | 0.458 | 0.483 | 0.329 | 0.486 | 0.457 | 0.363 | 0.472 | 0.547 |
| GR | 0.201 | 0.172 | 0.172 | 0.165 | 0.184 | 0.193 | 0.202 | 0.185 | 0.205 |
| IR | 0.201 | 0.167 | 0.367 | 0.164 | 0.248 | 0.314 | 0.200 | 0.228 | 0.412 |
| IT | 0.335 | 0.394 | 0.494 | 0.286 | 0.419 | 0.509 | 0.315 | 0.399 | 0.594 |
| MT | 0.011 | 0.004 | −0.022 | 0.001 | 0.011 | 0.006 | 0.006 | 0.006 | −0.008 |
| NL | 0.306 | 0.379 | 0.518 | 0.285 | 0.434 | 0.546 | 0.280 | 0.424 | 0.609 |
| PT | 0.154 | 0.264 | 0.548 | 0.098 | 0.282 | 0.541 | 0.160 | 0.260 | 0.646 |
| UK | 0.372 | 0.416 | 0.644 | 0.280 | 0.443 | 0.554 | 0.363 | 0.439 | 0.692 |
| BR | 0.442 | 0.767 | 0.767 | 0.408 | 0.735 | 0.638 | 0.461 | 0.766 | 0.803 |
| CN | 0.241 | 0.034 | −0.043 | 0.174 | 0.048 | −0.065 | 0.254 | 0.035 | −0.045 |
| IN | 0.060 | 0.158 | 0.159 | 0.088 | 0.145 | 0.043 | 0.070 | 0.148 | 0.122 |
| RU | 0.044 | 0.128 | 0.171 | 0.040 | 0.133 | 0.088 | 0.047 | 0.143 | 0.153 |
| ZA | 0.064 | 0.139 | 0.000 | 0.057 | 0.128 | −0.078 | 0.082 | 0.118 | 0.002 |
| BG | 0.023 | −0.063 | −0.116 | 0.031 | −0.071 | −0.098 | 0.020 | −0.082 | −0.089 |
| CZ | 0.179 | 0.028 | 0.356 | 0.160 | 0.044 | 0.407 | 0.179 | 0.037 | 0.387 |
| HU | 0.085 | 0.248 | 0.391 | 0.053 | 0.225 | 0.349 | 0.072 | 0.262 | 0.426 |
| PL | 0.154 | 0.131 | 0.390 | 0.122 | 0.128 | 0.414 | 0.142 | 0.158 | 0.428 |
| RO | −0.012 | 0.034 | 0.018 | 0.004 | 0.052 | −0.007 | −0.003 | 0.037 | −0.005 |
| EE | 0.119 | 0.018 | 0.071 | 0.091 | −0.006 | 0.086 | 0.125 | 0.019 | 0.038 |
| HR | 0.014 | 0.379 | 0.022 | −0.010 | 0.354 | 0.039 | 0.005 | 0.375 | 0.053 |
| LT | 0.014 | 0.129 | −0.046 | 0.025 | 0.102 | 0.029 | 0.007 | 0.128 | −0.007 |
| LV | 0.042 | 0.149 | 0.031 | 0.017 | 0.160 | 0.038 | 0.045 | 0.162 | 0.023 |
| DK | 0.148 | 0.260 | 0.204 | 0.100 | 0.276 | 0.163 | 0.135 | 0.272 | 0.181 |
| NO | 0.141 | 0.261 | 0.457 | 0.129 | 0.325 | 0.484 | 0.160 | 0.313 | 0.496 |
| SE | 0.318 | 0.418 | 0.398 | 0.275 | 0.466 | 0.432 | 0.309 | 0.451 | 0.467 |
| HK | 0.103 | 0.396 | 0.063 | 0.111 | 0.319 | 0.069 | 0.123 | 0.354 | 0.071 |
| JP | −0.025 | 0.108 | 0.001 | −0.036 | 0.099 | 0.059 | −0.034 | 0.083 | −0.001 |



**Table A3.** Cross correlations coefficients between lag predicted series during Pre-EZC. EZC and Post-EZC.

| Index | Germany | | | France | | | Italy | | |
|---|---|---|---|---|---|---|---|---|---|
| | Pre-Crisis | During Crisis | Post-Crisis | Pre-Crisis | During Crisis | Post-Crisis | Pre-Crisis | During Crisis | Post-Crisis |
| AT | 0.628 | 0.819 | 0.782 | 0.693 | 0.911 | 0.817 | 0.682 | 0.859 | 0.793 |
| BE | 0.715 | 0.745 | 0.878 | 0.873 | 0.944 | 0.927 | 0.804 | 0.910 | 0.851 |
| ES | 0.695 | 0.502 | 0.715 | 0.783 | 0.829 | 0.833 | 0.878 | 0.848 | 0.919 |
| FI | 0.643 | 0.909 | 0.781 | 0.727 | 0.937 | 0.782 | 0.639 | 0.896 | 0.671 |
| GR | 0.448 | 0.424 | 0.452 | 0.574 | 0.493 | 0.469 | 0.486 | 0.483 | 0.457 |
| IR | 0.466 | 0.567 | 0.652 | 0.576 | 0.814 | 0.678 | 0.458 | 0.766 | 0.718 |
| MT | 0.187 | 0.067 | −0.048 | −0.076 | 0.065 | −0.034 | −0.085 | 0.074 | −0.031 |
| NL | 0.702 | 0.793 | 0.906 | 0.837 | 0.955 | 0.919 | 0.816 | 0.898 | 0.764 |
| PT | 0.628 | 0.471 | 0.685 | 0.763 | 0.771 | 0.732 | 0.765 | 0.810 | 0.686 |
| UK | 0.907 | 0.841 | 0.829 | 0.883 | 0.867 | 0.823 | 0.785 | 0.796 | 0.642 |
| BR | 0.729 | 0.665 | 0.300 | 0.722 | 0.648 | 0.286 | 0.732 | 0.594 | 0.263 |
| CN | 0.350 | 0.017 | 0.579 | 0.154 | 0.172 | 0.492 | 0.173 | 0.129 | 0.289 |
| IN | 0.509 | 0.552 | 0.413 | 0.515 | 0.469 | 0.399 | 0.495 | 0.441 | 0.283 |
| RU | 0.323 | 0.638 | 0.213 | 0.306 | 0.654 | 0.174 | 0.262 | 0.580 | 0.135 |
| ZA | 0.400 | 0.574 | 0.483 | 0.436 | 0.644 | 0.464 | 0.467 | 0.587 | 0.370 |
| BG | 0.139 | 0.029 | −0.012 | 0.079 | 0.004 | 0.030 | 0.067 | 0.004 | 0.030 |
| CZ | 0.442 | 0.632 | 0.637 | 0.576 | 0.779 | 0.668 | 0.602 | 0.735 | 0.637 |
| HU | 0.276 | 0.574 | 0.455 | 0.382 | 0.803 | 0.436 | 0.426 | 0.794 | 0.425 |
| PL | 0.525 | 0.836 | 0.496 | 0.719 | 0.810 | 0.491 | 0.714 | 0.772 | 0.432 |
| RO | 0.137 | 0.289 | 0.100 | 0.130 | 0.560 | 0.115 | 0.089 | 0.533 | 0.113 |
| EE | 0.539 | 0.580 | 0.351 | 0.336 | 0.576 | 0.357 | 0.243 | 0.515 | 0.250 |
| HR | 0.324 | 0.116 | 0.062 | 0.230 | 0.121 | 0.108 | 0.273 | 0.120 | 0.105 |
| LT | 0.485 | 0.253 | 0.205 | 0.559 | 0.226 | 0.197 | 0.523 | 0.183 | 0.156 |
| LV | 0.037 | 0.024 | 0.099 | 0.091 | 0.226 | 0.071 | 0.087 | 0.195 | 0.002 |
| DK | 0.303 | 0.678 | 0.587 | 0.330 | 0.853 | 0.565 | 0.233 | 0.806 | 0.489 |
| NO | 0.603 | 0.801 | 0.688 | 0.555 | 0.865 | 0.688 | 0.600 | 0.806 | 0.577 |
| SE | 0.751 | 0.957 | 0.815 | 0.683 | 0.902 | 0.775 | 0.604 | 0.836 | 0.677 |
| HK | 0.527 | 0.794 | 0.531 | 0.512 | 0.623 | 0.497 | 0.461 | 0.560 | 0.354 |
| JP | 0.435 | 0.081 | 0.446 | 0.463 | 0.095 | 0.484 | 0.426 | 0.038 | 0.449 |
| DJI | 0.731 | 0.816 | 0.519 | 0.611 | 0.779 | 0.461 | 0.601 | 0.726 | 0.312 |
| NASDAQ | 0.756 | 0.778 | 0.508 | 0.601 | 0.756 | 0.465 | 0.609 | 0.704 | 0.348 |
| S&P500 | 0.790 | 0.798 | 0.580 | 0.693 | 0.764 | 0.537 | 0.705 | 0.712 | 0.379 |

**Table A4.** Cross correlations coefficients between lag residual series during Pre-EZC. EZC and Post-EZC.

| Index | Germany | | | France | | | Italy | | |
|---|---|---|---|---|---|---|---|---|---|
| | Pre-Crisis | During Crisis | Post-Crisis | Pre-Crisis | During Crisis | Post-Crisis | Pre-Crisis | During Crisis | Post-Crisis |
| AT | 0.457 | 0.668 | 0.688 | 0.547 | 0.787 | 0.723 | 0.611 | 0.720 | 0.691 |
| BE | 0.758 | 0.685 | 0.843 | 0.849 | 0.891 | 0.887 | 0.787 | 0.839 | 0.782 |
| ES | 0.650 | 0.517 | 0.683 | 0.566 | 0.831 | 0.814 | 0.752 | 0.883 | 0.903 |
| FI | 0.705 | 0.776 | 0.511 | 0.717 | 0.847 | 0.454 | 0.623 | 0.769 | 0.248 |
| GR | 0.337 | 0.270 | 0.326 | 0.417 | 0.324 | 0.359 | 0.252 | 0.312 | 0.390 |
| IR | 0.543 | 0.606 | 0.536 | 0.620 | 0.758 | 0.528 | 0.488 | 0.719 | 0.467 |
| MT | 0.002 | −0.064 | 0.000 | −0.024 | −0.046 | 0.004 | 0.030 | −0.042 | 0.013 |
| NL | 0.794 | 0.773 | 0.850 | 0.768 | 0.914 | 0.882 | 0.792 | 0.828 | 0.704 |
| PT | 0.638 | 0.483 | 0.592 | 0.618 | 0.786 | 0.657 | 0.625 | 0.812 | 0.641 |
| UK | 0.842 | 0.710 | 0.675 | 0.867 | 0.726 | 0.692 | 0.737 | 0.614 | 0.435 |
| BR | 0.659 | 0.440 | 0.121 | 0.603 | 0.417 | 0.129 | 0.660 | 0.303 | 0.105 |
| CN | 0.036 | 0.083 | 0.226 | 0.058 | 0.052 | 0.189 | 0.059 | −0.001 | 0.101 |
| IN | 0.139 | 0.297 | 0.292 | 0.297 | 0.342 | 0.314 | 0.218 | 0.274 | 0.200 |
| RU | 0.131 | 0.486 | 0.198 | 0.185 | 0.363 | 0.160 | 0.137 | 0.207 | 0.109 |
| ZA | 0.037 | −0.046 | 0.052 | −0.092 | −0.070 | 0.065 | −0.095 | −0.074 | 0.088 |
| BG | −0.021 | 0.019 | 0.051 | 0.068 | −0.011 | 0.061 | 0.157 | −0.019 | 0.061 |
| CZ | 0.278 | 0.452 | 0.484 | 0.378 | 0.575 | 0.481 | 0.455 | 0.514 | 0.443 |
| HU | 0.344 | 0.466 | 0.351 | 0.292 | 0.677 | 0.374 | 0.398 | 0.675 | 0.353 |
| PL | 0.384 | 0.673 | 0.440 | 0.503 | 0.639 | 0.445 | 0.480 | 0.573 | 0.371 |
| RO | 0.057 | 0.277 | 0.064 | 0.129 | 0.451 | 0.064 | 0.075 | 0.374 | 0.074 |
| EE | 0.383 | 0.382 | 0.209 | 0.353 | 0.372 | 0.226 | 0.161 | 0.252 | 0.144 |

**Table A4.** *Cont.*

| Index | Germany | | | France | | | Italy | | |
|---|---|---|---|---|---|---|---|---|---|
| | Pre-Crisis | During Crisis | Post-Crisis | Pre-Crisis | During Crisis | Post-Crisis | Pre-Crisis | During Crisis | Post-Crisis |
| HR | 0.036 | 0.076 | 0.183 | 0.047 | 0.073 | 0.211 | 0.265 | 0.024 | 0.193 |
| LT | 0.132 | 0.107 | 0.149 | 0.402 | 0.121 | 0.146 | 0.291 | 0.070 | 0.074 |
| LV | 0.025 | 0.130 | 0.050 | 0.070 | 0.173 | 0.036 | 0.009 | 0.168 | 0.015 |
| DK | 0.281 | 0.603 | 0.363 | 0.244 | 0.723 | 0.335 | 0.157 | 0.628 | 0.243 |
| NO | 0.575 | 0.695 | 0.482 | 0.409 | 0.725 | 0.511 | 0.565 | 0.662 | 0.353 |
| SE | 0.660 | 0.877 | 0.538 | 0.539 | 0.832 | 0.450 | 0.472 | 0.697 | 0.214 |
| HK | 0.229 | 0.255 | 0.256 | 0.319 | 0.237 | 0.283 | 0.166 | 0.088 | 0.193 |
| JP | 0.021 | 0.108 | 0.304 | 0.205 | 0.062 | 0.341 | 0.115 | 0.009 | 0.327 |
| DJI | 0.578 | 0.617 | 0.374 | 0.483 | 0.577 | 0.371 | 0.494 | 0.503 | 0.273 |
| NASDAQ | 0.562 | 0.580 | 0.361 | 0.457 | 0.554 | 0.350 | 0.509 | 0.464 | 0.265 |
| S&P500 | 0.647 | 0.594 | 0.405 | 0.547 | 0.556 | 0.408 | 0.594 | 0.473 | 0.303 |

**Table A5.** Country code and name of the country for the stock indexes.

| | Country Code | Country Name |
|---|---|---|
| 1. | AT | Austria |
| 2. | BE | Belgium |
| 3. | DE | Germany |
| 4. | ES | Spain |
| 5. | FR | France |
| 6. | GR | Greece |
| 7. | IR | Ireland |
| 8. | IT | Italy |
| 9. | MT | Malta |
| 10. | NL | The Netherlands |
| 11. | PT | Portugal |
| 12. | UK | United Kingdom |
| 13. | BR | Brazil |
| 14. | CN | China |
| 15. | IN | India |
| 16. | RU | Russia |
| 17. | ZA | South Africa |
| 18. | BG | Bulgaria |
| 19. | CZ | Czech Republic |
| 20. | HU | Hungary |
| 21. | PL | Poland |
| 22. | RO | Romania |
| 23. | EE | Estonia |
| 24. | HR | Croatia |
| 25. | LT | Lithuania |
| 26. | LV | Latvia |
| 27. | DK | Denmark |
| 28. | FI | Finland |
| 29. | NO | Norway |
| 30. | SE | Sweden |
| 31. | HK | Hong Kong |
| 32. | JP | Japan |
| 33. | DJI | US Dow Jones Index |
| 34. | NASDAQ | US NASDAQ |
| 35. | S&P500 | US Standard & Poor 500 |

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
