# Peer review of "Equity Market Contagion in Return Volatility during Euro Zone and Global Financial Crises: Evidence from FIMACH Model"

_jrfm, doi:10.3390/jrfm12020094_

Round 1

Reviewer 1 Report

The paper addresses an important and contemporary topic of the contagion between equity markets in various regions and time periods. The overall quality of the paper is good and the declared aim was reached. However, the paper requires some revisions in order to further increase its quality and be fully suitable for publication. I list my comments below in the approximate order of the paper.

The abstract stresses that 'squared' returns are used in the analysis - this issue is not covered in-depth in the remainder of the paper, it should be discussed carefully.

The abstract should include less presentation of the results and more information about the implications of the study.

The paragraph that starts on line 68 should be expanded, in particular with regard to the adopted approach based on the previous studies.

One of the main weaknesses of the paper is low quality of the presentation of the results. For example, in tables 1 and 2, authors use in some cases the full names of countries and in the other they use the names of the stock indexes (it should be consistent). The other issue is the unclear order of the names in both tables.

There are some redundant elements in the methodological section such as the equations of correlation coefficients. All equations require some revisions as their left sides are in the upper lines (they should be in the same line as the right sides).

Section 4 starts with the 'loss function' - it is rather unclear.

Authors focus on the approach discussed in the 2019 study yet other approaches should also be considered and compared.

Entire section 5 is poorly written which makes it difficult to follow and understand the results of the study. It should be rewritten. I suggest moving some of the tables to the appendix. Paragraph starting on the line 359 is very unclear and difficult to understand. Titles of figures 2 and 3 are identical, even though they cover different results.

Authors should write more about the implications of their study - it is an important missing aspect. Their results should also be compared to the conclusions of the previous studies.

Reference section requires some revisions. Another problem is references to some rather outdated publications or the ones that are not relevant in the studied context (e.g., Hurst 1951, 1956).

Author Response

Dear Editor,

I have attached a document in response to the reviewer.

Best,

Shahiduzzaman Quoreshi

Reviewer 2 Report

The paper is not acceptable for publication in its present form. However, it could be a very good paper if some major changes will be done. I have the following comments and suggestions, which can improve the quality of the paper:

1)The observation numbers for pre-global financial crises and post-Euro zone crisis are significantly more as compared with other periods. This difference may affect the estimation results.

2)Almost no new appropriate literature is cited. The only new literature cited by the authors is their own paper (Quoreshi and Mollah (2019)), which is forth coming. A huge number of literature on volatility of financial markets published in last two decades and related to this topic are ignored. The lines 8 and 9 even say that the topic is untouched, which is not right. Many literature using GARCH type models have addressed the topic in last two decades. To check, search in Google the keywords “volatility, financial markets, stock, GARCH, pdf”.

3)In cross-correlation test the squared values of (standardized) residuals could be used. The squares values of the returns are very similar to the squared values of the residuals, but they are not standardized.

4)The findings based on correlation of  squared returns can’t be a credible prove of relationship or impact. The cross-correlation function is more appropriate.

5)The cross-correlations coefficients without lags or leads should be also carefully considered and interpreted. Especially, when the impact is addressed.

6)The estimations including the cross-corelletaion coefficients with lags (Tables 4 to 7) are the most appropriate and important parts of this paper. The conclusion should be based on this part of the estimations.

7)The correlation could be constant or dynamic, conditional and unconditional. The authors have not check it.

Author Response

(The authors gave the same response as above.)

Round 2

Reviewer 2 Report

The revised version is acceptable for publication.

There is one point which seems inappropriate:

In lines 58-59 it is said that “However, none of these studies consider the contagion of squared stock returns as proxy for volatilities”, but the following paper which is cited more than 500 times has the same methodology in calculations, which rejects what the authors say.

“Cheung, Yin-Wong and Ng, Lilian K. 1996. “A causality-in-variance test and its application to financial markets prices.” Journal of Econometrics 72(1): 33-48.”

Author Response

I have now deleted the sentence.

Thanks!